# Intelligibility Sound Therapy Enhances the Ability of Speech-in-Noise Perception and Pre-Perceptual Neurophysiological Response

**DOI:** 10.3390/biology13121021

**Published:** 2024-12-06

**Authors:** Takashi Ishino, Kei Nakagawa, Fumiko Higashikawa, Sakura Hirokane, Rikuto Fujita, Chie Ishikawa, Tomohiro Kawasumi, Kota Takemoto, Takashi Oda, Manabu Nishida, Yuichiro Horibe, Nobuyuki Chikuie, Takayuki Taruya, Takao Hamamoto, Tsutomu Ueda, Louis Yuge, Sachio Takeno

**Affiliations:** 1Department of Otorhinolaryngology, Head and Neck Surgery, Graduate School of Biomedical Sciences, Hiroshima University, Kasumi 1-2-3, Minami-ku, Hiroshima 734-8551, Japan; sakura16@hiroshima-u.ac.jp (S.H.); rikto@hiroshima-u.ac.jp (R.F.); chie0324@hiroshima-u.ac.jp (C.I.); kwtm2022@hiroshima-u.ac.jp (T.K.); kota61@hiroshima-u.ac.jp (K.T.); odataka@hiroshima-u.ac.jp (T.O.); nm1027@hiroshima-u.ac.jp (M.N.); horibey@hiroshima-u.ac.jp (Y.H.); housejak@hiroshima-u.ac.jp (N.C.); ttaruya@hiroshima-u.ac.jp (T.T.); takao0320@hiroshima-u.ac.jp (T.H.); uedatsu@hiroshima-u.ac.jp (T.U.); takeno@hiroshima-u.ac.jp (S.T.); 2Department of Biomechanics, Graduate School of Biomedical and Health Sciences, Hiroshima University, Kasumi 1-2-3, Minami-ku, Hiroshima 734-8551, Japan; ryuge@hiroshima-u.ac.jp; 3Medical Center for Translational and Clinical Research, Hiroshima University Hospital, Hiroshima University, Kasumi 1-2-3, Minami-ku, Hiroshima 734-8551, Japan; fumiko@hiroshima-u.ac.jp

**Keywords:** intelligible-hearing (IH) sound, sensorineural hearing loss, speech-in-noise perception, cortical auditory-evoked fields (AEFs), N1m-P2m amplitude, magnetic mismatch negativity (MMNm)

## Abstract

Aural rehabilitation, including hearing aids, has been used for hearing disturbance such as hearing loss. However, there remains a need to investigate the effectiveness of aural rehabilitation and its method. In this study, we evaluated the potential of intelligible-hearing sound therapy as aural rehabilitation for hearing loss patients by adding the analysis of central cortex response by cortical auditory-evoked fields and magnetic mismatch negativity.

## 1. Introduction

Sensorineural hearing loss, including age-related, noise-induced, sudden, and genetic types, is associated with a lower quality of life due to difficulties in communication. Hearing loss impairs the ability to hear and understand speech, especially in background noise [1], and deteriorates speech recognition, leading to increased listening effort (LE) compared to individuals with normal hearing [2,3]. LE refers to the attentional requirements of cognitive resources required to comprehend auditory tasks [4,5]. Increased LE, along with additional requirements such as lip reading and top-down processing to understand conversations, further impairs listening-in-noise abilities [6,7] and causes fatigue [8,9,10,11]. The impact of fatigue, along with environmental and circumstantial factors, negatively affects cognitive processing, quality of life, and workplace productivity in individuals with hearing loss [4,9]. Additionally, sensorineural hearing loss like age-related hearing loss exacerbates speech comprehension difficulties due to the combined effects of degraded speech signal transmission from the cochlea to the central nervous system for cognitive and linguistic processing and declines in episodic memory, processing speed, and working memory. These sensory deficits from the peripheral to central nervous system accelerate cognitive and neural decline, often accompanied by fatigue. Regarding hearing disabilities, including fatigue induced by hearing loss, aural rehabilitation is an effective treatment. This encompasses hearing aids, hearing assistive technologies, auditory training, music therapy, counseling, and support groups. Aural rehabilitation can improve or mitigate cortical and cognitive function decline, as well as subcortical physiological deterioration [12]. Significant improvements in working memory, encompassing both cognitive and neural functions, have been demonstrated after six months of hearing aid use [12]. Additionally, improvements in short-term memory and learning ability have also been recognized after six months of hearing aid use [13]. Hearing aids are effective for tinnitus as they transmit high-frequency sounds that adequately stimulate the hearing pathway, leading to inhibition in the cortex and auditory pathway, which causes tinnitus. Neuro-music therapy is effective for recent-onset tinnitus [14], and constraint-induced sound therapy can prevent maladaptive auditory cortex reorganization in cases of sudden sensorineural hearing loss [15]. The effectiveness of aural rehabilitation, as mentioned above, makes hearing aids the most common treatment for mild-to-moderate hearing loss and tinnitus by amplifying deteriorated-frequency sounds. However, many people do not consistently use hearing aids [16,17,18]. Around 50% of users wear their hearing aids for less than 8 h per day, and about 40% do not use them daily [19]. Dissatisfaction with hearing aids often arises from various factors such as handling ability, sound quality, type of hearing aid, listening situation, level of experience, expectations, personality, and user attitude [20,21]. The key to successful rehabilitation is selecting aural rehabilitation methods that patients can maintain over the long term and the duration of rehabilitation. Considering aural rehabilitation compliance, other forms of aural rehabilitation besides hearing aids are also promising for hearing disturbances. Our previous report mentioned that rehabilitation with intelligible-hearing (IH) sound using an IH loudspeaker, which emphasizes high tone ranges and reduces sound distortion, can improve pronunciation discrimination by enhancing cortical auditory processing [22].

The mechanisms of aural rehabilitation primarily involve stimulating cortical auditory processing by supporting degraded hearing ability through the amplification of high-frequency sound ranges. Providing intelligible sound from a speaker or other rehabilitation system without causing dissatisfaction in its use will help hearing loss patients continue aural rehabilitation over the long term. Therefore, the concept of a personal communication support system (COMUOON pocket, Universal Sound Design, Tokyo, Japan) was developed based on the technology of the IH loudspeaker (a communication support system, COMUOON, Universal Sound Design, Tokyo, Japan). The sound collector emphasizes high-frequency sound ranges and reduces distortion of higher harmonics, similar to COMUOON. The improvement in hearing ability after using the COMUOON pocket is suggested by some patient-oriented self-assessment results in their daily lives.

To effectively measure the impact of aural rehabilitation, it is essential to monitor cortical function, which includes cognitive and neural functions, alongside physiological changes in auditory response. Cortical auditory-evoked potentials (AEPs) and auditory-evoked fields (AEFs) are long-latency potentials generated in response to auditory stimuli. The AEP is dominated by the P1-N1-P2 waveform complex, with P1 originating primarily in the lateral portion of Heschl’s gyrus [23,24], N1 in the primary and secondary auditory cortices [25,26,27,28], and P2 in multiple generators within the primary and secondary auditory cortex in the Heschl’s gyrus of both hemispheres [29]. AEPs/AEFs can be elicited by the onset, offset, or change in an ongoing sound and are generated without active participant attention or behavioral response [30]. AEPs/AEFs have been measured as objective markers for hearing, cognition, and various physical and mental illnesses [31,32,33].

Mismatch negativity (MMN) and magnetic MMN (MMNm) are pre-attentive auditory event-related potential (ERP) components elicited by infrequent changes in a repetitive acoustic pattern, reflecting the brain’s automatic response to auditory stimuli changes. MMN/MMNm is attributed to neural generators within the temporal and frontal lobes and represents higher-order processes in auditory deviance detection [34]. MMN/MMNm is used to assess the accuracy of central auditory processing, auditory discrimination abilities, and sensory memory. MMN/MMNm has been measured to evaluate the effectiveness of hearing aids and other auditory devices in speech-in-noise perception [35,36]. Therefore, the AEPs with the P1-N1-P2 waveform complex and MMN/MMNm have been used to detect the response of the auditory cortex, demonstrating the effectiveness of aural rehabilitation.

As improvements in hearing ability in daily life after using the COMUOON pocket are occasionally recognized, we hypothesized that aural rehabilitation with IH sound in hearing loss patients can decrease the attentional requirements of cognitive resources by amplifying deteriorated-frequency sounds. This can alter the response of cortical auditory processing, thereby improving auditory discrimination ability in conversations with or without noise. Given the potential of aural rehabilitation with IH sound to enhance discrimination ability in ordinary conversations with or without environmental noise, the aim of this study was to evaluate the relationships between aural rehabilitation with IH sound for hearing loss patients and improvements in hearing level, speech perception ability, self-assessment of hearing loss, and various hearing abilities associated with auditory processing. This evaluation included AEF and MMNm analysis to assess changes in auditory processing.

## 2. Materials and Methods

### 2.1. Participants

Participants underwent pure-tone audiometric tests (PTA) to measure both air conduction (125, 250, 500, 1000, 2000, 3000, 4000, 8000, 10,000, and 12,000 Hz) and bone conduction (250, 500, 1000, 2000, 3000, and 4000 Hz) thresholds. Adult participants with hearing impairment, native Japanese speakers (17 males and 23 females, 68.43 ± 9.23 years) who showed thresholds over 30 dB at the above frequencies in either ear, were recruited for this study. They had no history of neurological or psychiatric diseases and were not current hearing aid users (3 participants had previously used hearing aids for 1 to 8 years, with a period of 1 to 7 years since last use). None of the participants were musicians or had received any special musical training.

The study protocol was approved by the Ethics Committee of Hiroshima University. All participants provided written consent prior to participation and received compensation for the study.

### 2.2. Study Design

All participants underwent cognitive evaluation using the Mini-Mental State Examination Japanese (MMSE-J: cutoff value ≤ 23 for dementia, ≤27 for mild cognitive impairment) on day 1. Participants in this prospective study were fitted with earphone-type sound collectors. Normal sound therapy using a standard sound collector was provided to participants randomly assigned to the control group (Ctrl). The IH sound collector (COMUOON pocket, Universal Sound Design, Tokyo, Japan), which provides high-quality sound by emphasizing high-frequency ranges and reducing distortion of higher harmonics for better formant perception, was provided to the other participants as the IH sound therapy group (HQ). Both groups received sound therapy through the sound collectors by listening to preferred songs and/or tunes selected by each participant for one hour a day for 35 days. Self-evaluation questionnaires for hearing problems related to voice (Table 1) [37], auditory processing tests (APTs) including the gap detection test (GDT), fast speech test (FST), and speech-in-noise test (SINT: signal-to-noise ratio (S/N) +10, +5, 0, −5, −10, −15), PTA as described above, and speech perception tests using a Japanese 67-S were conducted on days 1 and 35. The better hearing ear was defined using a four-frequency pure-tone average (PTA) at the thresholds of 500, 1000, 2000, and 4000 Hz [38].

#### 2.2.1. GDT

The GDT was utilized to assess temporal processing ability. Each trial included three presentations of white noise without a silent gap and one presentation with a silent gap, with durations ranging from 2 to 34 ms. Participants responded to the presence of a gap in each trial. The stimuli were adaptively varied, and the threshold of gap duration yielding a 50% correct performance was measured.

#### 2.2.2. FST

The FST was employed to evaluate auditory closure ability. The stimuli comprised 20 context-free Japanese three-clause sentences, delivered at three different speaking speeds: normal, 1.5×, and 2×. Participants were required to respond to the sentences at each speed. The percentages of correct responses were calculated for the beginning, middle, and end of the clauses at each speaking speed.

#### 2.2.3. SINT

The SINT was utilized to assess auditory closure ability. The stimuli consisted of a mixture of 36 two-mora Japanese words and speech-spectrum noise, with signal-to-noise ratios (S/N) of −15, −10, −5, 0, +5, and +10 dB. Participants responded to the presented words, which were simultaneously and randomly delivered to both ears. The accuracy rates of the SINT at each S/R were calculated.

### 2.3. Magnetoencephalography Recording

Magnetoencephalography (MEG) recordings were conducted in a magnetically shielded room (MSR). Due to equipment repairs, 27 recruited participants were examined (Table 1). Functional auditory-evoked fields (AEFs) and MMNm were measured using a 306-channel MEG system (Vectorview; ELEKTA Neuromag, Helsinki, Finland) with 102 identical triple-sensor elements: two orthogonal planar gradiometers and one magnetometer coupled to a multi-superconducting quantum interference device (SQUID), providing three independent measurements of the magnetic fields. Data were recorded with a band-pass filter of 0.1–200 Hz and digitized at 1000 Hz. Epochs with MEG signals larger than 3.0 pT/cm were excluded from averaging. Collected MEG signals from 204-channel planar-type gradiometers were analyzed with an offline band-pass filter of 1–50 Hz.

#### 2.3.1. Auditory-Evoked Fields (AEFs)

A train of clicks (60 ms) was presented as the auditory stimulus, consisting of single sine waves of 1 ms at 1000 Hz and 4000 Hz, with a repetitive frequency of 3.33 Hz, at three different sound pressures (50 dB, 55 dB, and 60 dB HL), with 240 repeats of the same click. Auditory stimuli were presented bilaterally, respectively. Each stimulus started at 50 dB HL and increased in steps of 55 dB and 60 dB HL. Additional bilateral auditory stimuli (1000 Hz, 60 dBHL) were also presented to identify auditory cortical regions. Sound stimuli were delivered through plastic tubes and earpieces (E-A-RTone 3A; Aero Company, Indianapolis, IN, USA) using the NBS Presentation software (version 24.0) (Neurobehavioral Systems, Albany, CA, USA).

An equivalent current dipole (ECD) analysis was applied using Neuromag source modeling software (version 5.5.20) (ELEKTA Neuromag, Helsinki, Finland). The ECD model of magnetic N1 (N1m) was estimated using 14–20 sensors (7–10 pairs of gradiometers) in the bilateral hemispheres with bilateral auditory stimulation (1000 Hz, 60 dBHL). The ECD coordinates estimated in the bilateral superior temporal gyrus (STG) based on a goodness of fit (GOF) exceeding 80% were selected. The ECD coordinates were superimposed on the responses recorded with the other conditions. The peak-to-peak amplitude of N1m-P2m (magnetic P2) in each task was analyzed before and after the sound therapy.

#### 2.3.2. MMNm

The oddball task was presented bilaterally to the ears with two random auditory stimuli: a deviant stimulus (20%, 1000 Hz) and a standard stimulus (80%, 500 Hz), both with a repetitive frequency of 3.33 Hz and a sound pressure of 50 dB HL. The response was analyzed in the same manner as in the AEF analysis. MMNm responses were computed by subtracting the evoked response for the standard stimulus from that of the deviant stimulus, and the peak-to-peak amplitude around 150 ms was analyzed before and after the sound therapy.

### 2.4. Statistical Analysis

Experimental data were analyzed using linear mixed-effects (LME) models utilizing the lme4 package (version 1.1-35.5) and the lmerTest package (version 3.1-3) in R. The significance was further adjusted for multiple comparisons using the Bonferroni approach. All *p*-values were two-tailed, with unadjusted *p*-values less than 0.05 considered significant, and participants were included as a “by-subject” random effect in the linear mixed-effect model. For the GDT, FST, SINT, and MEG data, the LME model was administrated to the groups HQ, Ctrl, HF-HQ, and HF-Ctrl.

## 3. Results

### 3.1. The Participants

Table 2 and Figure 1 exhibit patient demographics and audiograms. There were no significant differences in the demographics of participants between the two sound therapy groups. The IH sound collector primarily emphasizes high-frequency ranges and reduces distortion of higher harmonics to improve formant perception. Therefore, we created subcategory groups, excluding those with a threshold level over 70 dB at 10,000 Hz bilaterally (HF-Ctrl: high-frequency hearing loss control/HF-HQ: high-frequency hearing loss HQ), to analyze the effectiveness of the IH sound therapy. The maximum output level of the IH sound collector is approximately 75–80 dB HL in high-frequency ranges around 10,000 Hz. The better hearing ear was defined based on the average threshold level according to WHO criteria, averaging thresholds at 500, 1000, 2000, and 4000 Hz. All PTA, SPT, and auditory processing tests were analyzed in terms of the better or worse hearing ear based on the PTA results from day 1.

### 3.2. Self-Evaluation Questionnai

The self-evaluation questionnaire for hearing problems related to voice did not show any differences among the sound therapy groups, including HF-Ctrl and HF-HQ, on days 1 and 35, nor in the changes between days 1 and 35.

### 3.3. PTA Measurement and Speech Perception Test

The average threshold level in air conduction PTA, the sound levels at 50% perception, and the maximum perception percentage in SPT did not show any differences among the sound therapy groups, including HF-Ctrl and HF-HQ, on days 1 and 35, nor in the changes between days 1 and 35.

### 3.4. Auditory Processing Test

#### 3.4.1. GDT

The GDT did not show any differences among the sound therapy groups, including HF-Ctrl and HF-HQ, on days 1 and 35, nor in the changes between days 1 and 35 (Table 3).

#### 3.4.2. FST

The FST did not show any differences among the sound therapy groups, including HF-Ctrl and HF-HQ, on days 1 and 35, nor in the changes between days 1 and 35 (Table 3).

#### 3.4.3. SINT

The SINT results showed that the accuracy rate exceeded 50% only for S/N +10, +5, and +0 in both pre- and post-therapy conditions. In the groups with an accuracy rate over 50%, the SINT with S/N +10 in the better hearing ear showed significant improvement after sound therapy in the HQ and HF-HQ groups (Figure 2). In contrast, the Ctrl and HF-Ctrl groups did not exhibit any significant differences. Additionally, the improved accuracy rate in HQ and HF-HQ, calculated by subtracting the pre-therapy accuracy rate from the post-therapy accuracy rate, showed a significantly higher improvement rate in HF-HQ compared to HF-Ctrl in the SINT with S/N +10 in the better hearing ear. The improvement in the SINT with S/N +10 after sound therapy was widely recognized in HQ, independent of the hearing level in the better hearing ear and the accuracy rate with S/N +10 before sound therapy.

### 3.5. Magnetoencephalography Recording

#### 3.5.1. ECD Analysis Moment and Auditory-Evoked Potentials

Bilateral auditory stimuli of 1000 Hz at three different sound pressures (50 dB, 55 dB, and 60 dB HL) elicited clear magnetic deflections (N1m and P2m) after stimulus onset in the bilateral STG in response to each stimulus side. However, auditory stimuli of 4000 Hz at the same sound pressures did not show clear magnetic deflections of N1m and P2m under any conditions on day 1 (Figure 3). Based on these results, we analyzed the N1m and P2m components in the auditory stimuli of 1000 Hz. Given that the SINT with S/N +10 in the better hearing ear in HQ and HF-HQ showed significant improvement after sound therapy, we analyzed AEFs and N1m-P2m peak to peak in the left STG, between pre- and post-sound therapy. Inter-peak amplitudes of N1m-P2m were measured by subtracting the instantaneous amplitude at the peak latency of P2m from the instantaneous amplitude at the peak latency of N1m. Compared to Ctrl, HQ showed a significant increase in the amplitude of N1m-P2m in the left STG with stimuli of 55 and 60 dB from the better hearing ear after sound therapy (Figure 4). The latency and amplitude of N1m did not show any significant differences in any group in this study in either STG.

#### 3.5.2. MMNm

Superimposed auditory-evoked fields in HQ and Ctrl are shown in Figure 5. Clear deflections (MMNm) were observed around 100–250 ms. MMNm was measured and analyzed between HQ/HF-HQ and Ctrl/HF-Ctrl. In HQ, a significant increase in MMNm amplitude was observed at the left STG after sound therapy, whereas no significant differences were observed at the right STG in HQ/HF-HQ or at either STG in HF-Ctrl. In HF-HQ, the tendency of an increase in MMNm amplitude was observed at the left STG after sound therapy, but there was not a significance (Figure 6).

#### 3.5.3. Inter-Peak Amplitude of N1m-P2m and MMNm

The percentage of subjects showing improvement in MMNm amplitude in the left STG and the inter-peak amplitude of N1m-P2m in the left STG at a stimulation sound level of 55 dB from the better hearing ear is shown in Table 4. The HQ and HF-HQ groups exhibited a higher improvement rate compared to the Ctrl and HF-Ctrl groups in the inter-peak amplitude of N1m-P2m and MMNm amplitudes in the left STG. In the HQ and HF-HQ groups, improvement in either the inter-peak amplitude of N1m-P2m or MMNm amplitudes was 90–93%, and improvement in both was 70–71%, compared to the Ctrl and HF-Ctrl groups, which showed 63–69% and 13–23% improvement, respectively.

#### 3.5.4. LME Model for SINT with S/N+10

Focusing on the significance of the inter-peak amplitude of N1m-P2m in the left STG from better hearing ear stimulation and MMNm amplitude in HQ and HF-HQ, linear mixed-effects models were applied for detecting the most important factor for the SINT with S/N +10 in the better hearing ear in the IH sound therapy group. In this analysis, participants were included as a “by-subject” random effect, and IH sound therapy, age, MMSE, PTA, hearing ability under the threshold level of 70 dB at 10,000 Hz, inter-peak amplitude of N1m-P2m in the left STG at 55 dB from the better hearing ear, and MMNm amplitude in the left STG were set as fixed effects. Linear mixed-effects regression analysis revealed that hearing ability under the threshold level of 70 dB at 10,000 Hz (*p* < 0.01) and IH sound therapy (*p* < 0.05) were significant predictors of the outcome variable of the SINT with S/N+10 in the better hearing ear. On the other hand, inter-peak amplitude of N1m-P2m and MMNm amplitudes did not show any significance in the LME model (Table 5).

## 4. Discussion

The effectiveness of aural rehabilitation for hearing disturbances, such as hearing loss and tinnitus, is widely recognized. Hearing aids, a common method of aural rehabilitation, are extensively used worldwide, especially for patients with hearing loss. Sensory devices like hearing aids can mitigate the effects of sensory loss by restoring certain aspects of sensory functioning [39,40]. Hearing aids can improve speech-in-noise perception, which is linked to cognitive and signal processing [41]. Additionally, the use of hearing aids enhances working memory performance, which is often diminished by hearing loss, and increases cortical amplitudes, correlating with changes in working memory [12]. These findings suggest that aural rehabilitation for sensorineural hearing loss can improve cognitive and neural functions by restoring the sensory–cognitive connection, thereby enhancing speech-in-noise perception through improved working memory performance. In this study, IH sound therapy improved the SINT of S/N +10 in the better hearing ear, with significant improvement also observed in HF-HQ compared to HF-Ctrl. The effectiveness of IH sound therapy was evident in hearing loss up to around 60 dB, and the improved accuracy rate appeared to be inversely proportional to the pre-accuracy rate of the SINT of S/N +10. These results suggest that IH sound therapy is as effective in improving hearing function as hearing aids. The IH sound collector can amplify sound up to approximately 75–80 dB at around 10,000 Hz (company-provided data; and IH sound therapy was particularly effective for patients with sensorineural hearing loss who can hear high-frequency sounds at 10,000 Hz at 70 dB. Adequate intelligibility stimulation in the high-frequency range is necessary to mitigate the effects of sensory hearing loss and improve speech-in-noise perception in these patients. Improvement in the SINT of S/N +10 in the better hearing ear was also observed after normal sound therapy in some Ctrl and HF-Ctrl cases. Patients who cannot hear the threshold level of 70 dB at 10,000 Hz cannot receive high-frequency sound stimulation from the sound therapy due to limitations in both cochlear function and sound collector specifications, but they can still receive adequate low-frequency sound stimulation.

Based on the improved accuracy rates between Ctrl and HF-Ctrl, the enhancement provided by standard sound therapy was primarily observed in patients with severe high-frequency hearing loss, specifically those with thresholds exceeding 70 dB at 10,000 Hz. The results suggest that some patients with severe sensorineural hearing loss at 10,000 Hz can benefit from low-frequency stimulation without intelligibility, as it effectively stimulates the auditory system. These findings and the LME model indicate that early-stage sensorineural hearing loss (up to 70 dB at 10,000 Hz) requires high-frequency intelligibility sound stimulation to restore the sensory–cognitive connection. In contrast, late-stage sensorineural hearing loss (over 70 dB at 10,000 Hz) benefits more from low-frequency sound stimulation due to more severe high-frequency hearing loss than early-stage sensorineural hearing loss. Sound therapy for severe high-frequency hearing loss may improve speech-in-noise perception even without intelligibility sound. This is likely because sensorineural hearing loss typically begins in the high-frequency range and progresses to lower frequencies. In the early stages, patients lack high-frequency information and need high-frequency intelligibility sound. In the late stages, they require a full range of frequency information but cannot receive adequate high-frequency stimulation due to cochlear limitations and sound collector specifications. It is recognized that most patients with high-frequency hearing loss can utilize low-frequency speech cues better than those with normal hearing, as the brain areas usually responsive to high frequencies adapt to analyze low-frequency sounds [42]. In this context, some patients can reestablish the sensory–cognitive connection even with low-quality sound, as it still provides some stimulation to the central cortex. Currently, the degree of distorted tonotopy is identified as a major factor contributing to degraded speech intelligibility, particularly in speech-in-noise perception [43,44]. Distorted tonotopy increases across-fiber responses in the auditory nerve and reduces the information capacity to the brain. The impact of distorted tonotopy on individual differences may vary depending on the etiology. It causes interruptions in high-frequency regions of speech coding, leading to hearing disturbances in speech-shaped noise caused by multiple talkers [43]. Our results showed that the improvement rate of speech-in-noise perception varied from poor to good with non-IH sound therapy and from fair to good with IH sound therapy. This indicates that non-IH sound therapy can sometimes offer better stimulation for speech-in-noise perception but can also occasionally worsen it due to the lack of intelligibility sound, which may fail to stimulate or even deteriorate the sensory–cognitive connection. The difference between these sound therapies may stem from the underlying pathophysiology of distorted tonotopy, as it disrupts auditory nerve fiber responses at high-frequency regions with low-frequency stimulus energy [44]. Non-IH sound therapy may primarily provide low-frequency enhanced sound compared to IH sound therapy, causing disruption in auditory nerve fiber responses in cases of poor response. In contrast, IH sound therapy consistently provides better stimulation with high-frequency enhanced intelligible sound, effectively stimulating the sensory–cognitive connection in all hearing loss patients. The stimulation algorithm is similar to hearing aid algorithms that prescribe more gain at high frequencies, counteracting the effects of distorted tonotopy [45]. Therefore, IH sound therapy is superior to non-IH sound therapy and is recommended for all patients with sensorineural hearing loss.

However, it remains unknown whether aural rehabilitation using IH sound therapy can counteract the deterioration of brain function associated with auditory processing. The N1m-P2m complex, a component of the AEFs, originates primarily in the auditory cortex and is associated with the detection and processing of auditory stimuli [29,46]. The amplitude and latency of these waves can vary depending on cognitive load and the individual’s working memory capacity, reflecting the brain’s increased effort to process and store auditory information [46,47]. The N1-P2 complex is also a marker of sensory memory [48] and auditory learning plasticity [49,50,51]. The N1-P2 response has advantages such as better frequency specificity, inclusion of higher neurologic centers, and less reliance on patient relaxation [52]. Consequently, the N1-P2 complex has been studied in various contexts, including its relationship with cognitive processes like working memory [53,54]. MMN/MMNm is also measured to evaluate automatic cortical auditory processing, selective attention, and neuronal memory traces [55,56]. MMN/MMNm depends on the degree of sensorineural hearing loss and the intensity of stimuli (sound volumes) [57], with significantly reduced amplitude and prolonged latency in hearing-impaired individuals compared to those with normal hearing [58,59]. Music training can shorten the latencies of harmonic, complex, and speech stimuli, possibly facilitating neural processing of acoustic parameters, which manifests in concomitant behaviors like speech perception [60]. The shortening of MMN/MMNm latencies and increased MMN/MMNm amplitude by speech stimuli were also observed after using IH loudspeakers for IH sound stimulation, which is the basis of this study. Therefore, MMN is measured to evaluate hearing levels and discriminative functions, including speech perception, and to analyze the effectiveness of aural rehabilitation [22]. In our study, HQ exhibited significant improvement in N1m-P2m amplitude at 55 and 60 dB and HF-HQ exhibited nearly significant improvement at 60 dB in the left superior temporal gyrus (Lt STG). Additionally, HQ showed a significant increase and HF-HQ showed a nearly significant increase in MMNm amplitude in the left STG after IH sound therapy. This proves that repetitive stimulation of the auditory cortex can enhance auditory processing [61]. The findings indicate that IH sound therapy provides superior sound stimulation, improving cognitive processes, including working memory and auditory learning plasticity. Given that MMN/MMNm amplitude reflects automatic cortical auditory processing, selective attention, and neuronal memory trace, and is closely associated with the enhancement of discriminative functions such as speech perception regardless of effort, these results explain the improved SINT scores in S/N+10 in the better hearing ear following IH sound therapy.

The LME model for the SINT scores in S/N +10 in the better hearing ear revealed that both IH sound therapy and hearing ability under the threshold level of 70 dB at 10,000 Hz were significant predictive factors for the improvement in the SINT scores in S/N +10, suggesting that hearing ability under the threshold level of 70 dB at 10,000 Hz may be the best indicator for IH sound therapy. The results also showed that the effectiveness of IH sound therapy did not correlate with the inter-peak amplitude of N1m-P2m and MMNm amplitudes in the left STG. Given that the improved rate of both brain responses in IH sound therapy were about 80% individually, about 90% in either, and about 70% in both, it is suggested that the effectiveness of IH sound therapy could not express a correlation with these brain responses because most participants in the IH sound therapy group showed improvement in brain responses regardless of the improvement in the SINT. These findings suggest that the effectiveness of IH sound therapy may initially lead to improvements in brain responses, followed by improvements in speech-in-noise perception.

The improvement in speech-in-noise perception is likely induced by aural rehabilitation using IH sound therapy, with sound being a key factor in stimulating the auditory cortex and enhancing cognitive processes, including working memory. Furthermore, the effectiveness of sound therapy with music has already been demonstrated. Previous reports have indicated that constrained listening rehabilitation with music, in addition to corticosteroid therapy, can significantly improve both hearing levels and auditory cortex responses in cases of sudden sensorineural hearing loss [15]. In our study, we used music for sound therapy, which may also contribute to the positive results in speech-in-noise perception in the better hearing ear. Hearing impairment induces a decline in speech perception, especially in noisy situations, leading to poor communicative ability and a decline in cognitive function, which often complicates other dysfunctions such as dementia [62]. Therefore, various aural rehabilitation methods have been investigated, but no unified method has been established [63,64]. On the other hand, hearing aids can improve speech-in-noise perception by enhancing cognitive function through changes in the central cortex response. The development of hearing aids has focused on device modifications to reduce user dissatisfaction, but some individuals persistently refuse to use them. IH sound therapy with repetitive music stimuli also shows potential for improving speech-in-noise perception, similar to hearing aids. The duration of rehabilitation is a crucial aspect of aural rehabilitation. IH sound therapy with repetitive music stimuli is likely more enjoyable than simply hearing natural sounds, and the sound collector provides only music without amplifying background environmental noise. Therefore, improved sound intelligibility mixed with music may become an alternative method for hearing-impaired patients instead of hearing aids. A long-term study is needed to further elucidate the effectiveness of IH sound therapy.

However, there are some limitations. First, all HQ therapy groups had significantly improved results in the SINT in the better hearing ear after the sound therapy and no improvement was exhibited in all Ctrl groups, but all Ctrl pre-therapy groups showing a higher accuracy rate compared to the HQ pre-therapy group may enhance the results. Second, we could not analyze the optimal periods for IH sound therapy, such as the duration of daily use, the total time of sound therapy, and the long-term effects of IH sound therapy. Third, we could not include more severe hearing loss patients to determine whether they could be candidates for IH sound therapy, and we did not assess satisfaction with IH sound therapy compared to hearing aids. Fourth, these aspects should be evaluated for IH sound therapy to become a more common aural rehabilitation method in the future. Further studies are needed to investigate the efficacy and satisfaction of IH sound therapy for sensorineural hearing loss.

## 5. Conclusions

IH sound therapy can improve speech-in-noise perception in the better hearing ear, as demonstrated by AEF and MMNm results, which reflect enhanced central cortex response and working memory ability. Therefore, IH sound therapy can be a valuable aural rehabilitation method for sensorineural hearing loss, comparable to hearing aids, especially in patients with hearing ability under the threshold level of 70 dB at 10,000 Hz.

## Figures and Tables

**Figure 1 biology-13-01021-f001:**
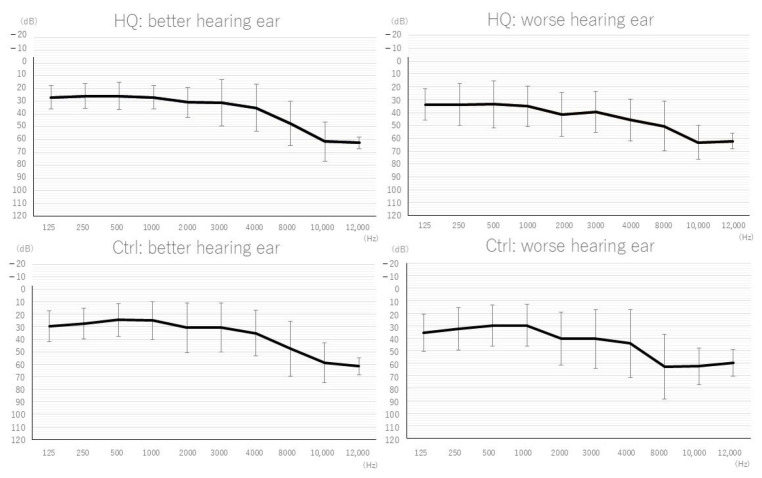
Patients’ audiograms. Both better and worse hearing ear in HQ and Ctrl showed mainly high-frequency hearing loss and no significance between HQ and Ctrl (mean ± SD).

**Figure 2 biology-13-01021-f002:**
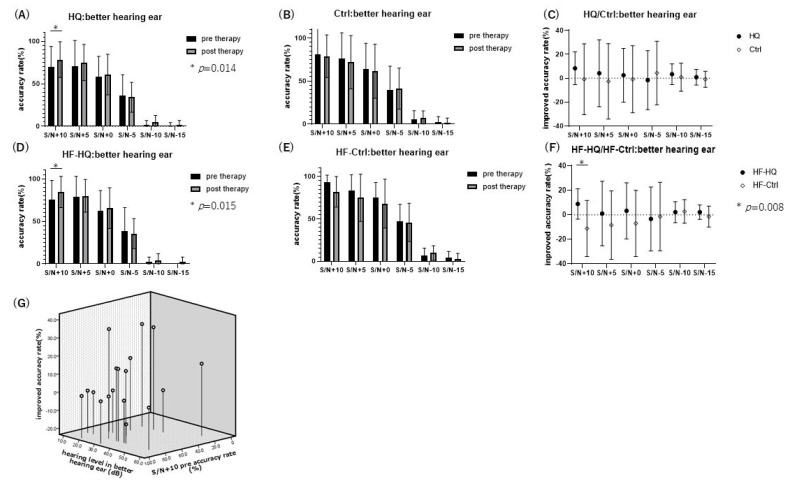
Speech-in-noise test (SINT) in the better hearing ear: HQ and HF-HQ exhibited significant improvement in S/N +10 after sound therapy (**A**,**D**), whereas Ctrl and HF-Ctrl did not show any significant differences (**B**,**E**). The improved accuracy rate in the better hearing ear in HQ did not show any significant differences compared to Ctrl (**C**), but HF-HQ exhibited significant improvement in S/N +10 compared to HF-Ctrl (**F**) with Bonferroni correction (adjusted *p*-value = 0.017, where the significance threshold was 0.05/3 = 0.0167). (**G**) Hearing level in the better hearing ear and improved accuracy rate with the accuracy rate in S/N +10 before sound therapy in HQ. Improvement in SINT in S/N +10 was observed across all hearing levels in this study, with the higher improvement seen in those with low accuracy rates before sound therapy (*: significance).

**Figure 3 biology-13-01021-f003:**
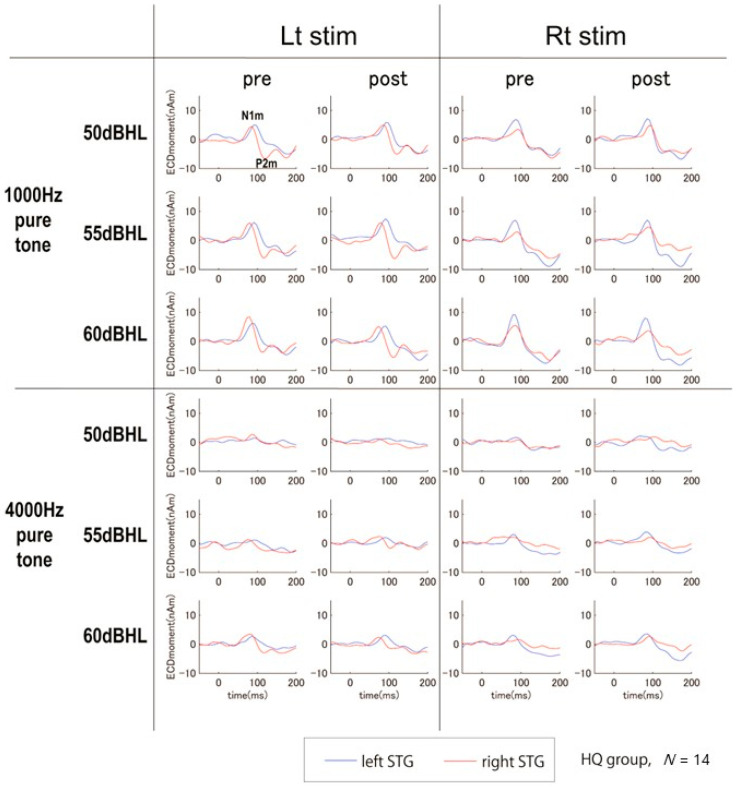
STG responses to 1000 Hz and 4000 Hz pure-tone stimulation at 50, 55, and 60 dB (HQ group, *N* = 14) (Lt: left, Rt: right): Stimulation at 1000 Hz elicited N1m and P2m responses, whereas stimulation at 4000 Hz did not elicit any clear magnetic deflections.

**Figure 4 biology-13-01021-f004:**
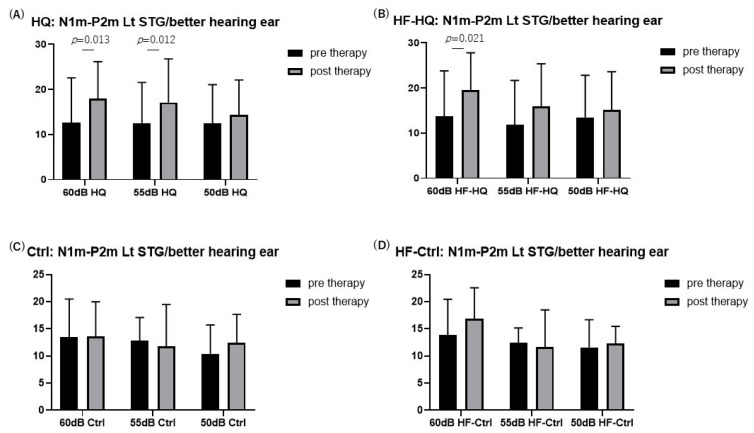
N1m-P2m amplitude in the left STG (Lt STG): HQ exhibited significant improvement in N1m-P2m amplitude at 55 and 60 dB, whereas Ctrl did not show any significant change in the left STG after sound therapy. HF-HQ exhibited improvement in N1m-P2m amplitude at 60 dB, but there was no significance after Bonferroni correction (adjusted *p*-value = 0.017, where the significance threshold was 0.05/3 = 0.0167 compared to the unadjusted *p*-value = 0.021) (*: significance).

**Figure 5 biology-13-01021-f005:**
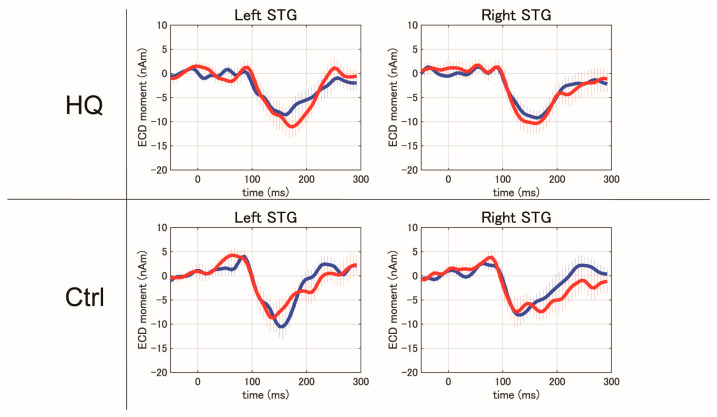
Superimposed auditory-evoked fields in HQ and Ctrl. Blue indicates pre-sound therapy, and red indicates post-sound therapy. Clear deflections (MMNm) were observed around 100–250 ms in both STG for HQ and Ctrl. Data are represented as mean ± SD.

**Figure 6 biology-13-01021-f006:**
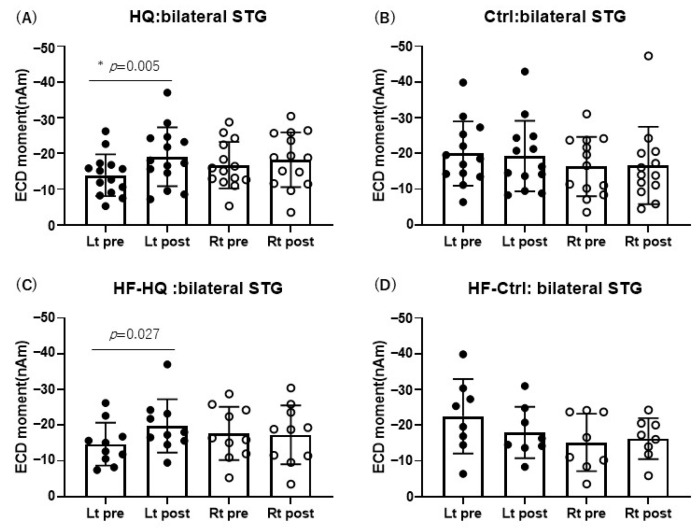
MMNm amplitude in left (Lt) and right (Rt) STG at pre- and post-sound therapy: HQ showed a significant increase in MMNm amplitude at the left STG after sound therapy, whereas Ctrl/HF-Ctrl did not exhibit any significant differences between pre- and post-sound therapy. In HF-HQ, the tendency of an increase in MMNm amplitude was observed at the left STG after sound therapy, but there was no significance after Bonferroni correction (adjusted *p*-value = 0.025, where the significance threshold was 0.05/2 = 0.025, compared to the unadjusted *p*-value = 0.027) (*: significance).

**Table 1 biology-13-01021-t001:** Self-evaluation questionnaire for hearing problems related to voice. The total score was calculated. The translated version of the test is shown above and the original test used in this study is shown below.

Self-Evaluation Questionnaire for Hearing Problems	Yes (×5)	Maybe (×3)	No (×1)
When multiple people talk at once, it’s hard to understand.	□	□	□
I often can’t understand conversations in the car.	□	□	□
Recently, it feels like people around me are mumbling and not speaking clearly.	□	□	□
My family and friends seem to think I should try wearing a hearing aid.	□	□	□
I work (or have worked) in a noisy environment with loud sounds.	□	□	□
I ask the other person to repeat themselves, or I guess what they said.	□	□	□
I find it easier to understand when I look at the person I’m talking to.	□	□	□
I often can’t understand the dialogue in TV shows and dramas.	□	□	□
Sometimes I don’t notice my phone ringing.	□	□	□
“I struggle to understand the content of discussions in meetings, classes, and gatherings.	□	□	□
	Total		Point
**聞こえの自己評価法**	**そうだ (×5)**	**そうかもしれない (×3)**	**そんなことはない (×1)**
二人以上の人が同時に話し始めるとよく聞き取れなくなる	□	□	□
自動車の中での話が、何を言っているのかよく聞き取れない	□	□	□
このごろ、周囲の人がモグモグとはっきりした声で話していないように感じる	□	□	□
家族や知人などが、補聴器をつけてみたらと考えているようだ	□	□	□
騒音の多い職場やうるさい大きな音のする環境にいる (いたことがある)	□	□	□
相手にもう一度繰り返し言って欲しいと頼んだり、そうでなければ推測して判断する	□	□	□
話し相手の顔見ているほうが話がよく分かると感じる	□	□	□
テレビ・ドラマの中のセリフがよく聞き取れない	□	□	□
携帯電話の呼び出し音に気が付かないことがある	□	□	□
集会、会議、授業などの場での話の内容がわからなくて困る	□	□	□
	計		点

**Table 2 biology-13-01021-t002:** Patients’ demographics. There were no significant differences in the demographics of participants between the two sound therapy groups. Better/worse ear PTA: pure-tone audiometry in better/worse hearing ear, MMSE: Mini-Mental State Examination Japanese, AEFs: auditory-evoked fields, MMNm: magnetic MMN, HQ: IH sound therapy group, Ctrl: control, HF-HQ: high-frequency hearing loss HQ, HF-Ctrl: high-frequency hearing loss control. Data shows mean ± SD.

Audiological Examination	HQ	Ctrl	*p* Value
Case	20	20	
Sex	M10, F10	M7, F13	*p* = 0.337
Age	69.4 ± 7.81	67.45 ± 10.57	*p* = 0.511
Better ear PTA	29.5625 ± 11.314	28.8125 ± 15.290	*p* = 0.861
Worse ear PTA	38.9375 ± 14.936	36.1250 ± 18.987	*p* = 0.606
MMSE	28.9 ± 1.293	28.45 ± 2.0894	*p* = 0.418
**Audiological examination**	**HF-HQ**	**HF-Ctrl**	***p* value**
Case	15	13	
Sex	M8, F7	M9, F4	*p* = 0.229
Age	69.47 ± 7.5	64.85 ± 10.645	*p* = 0.192
Better ear PTA	28.9167 ± 12.772	22.0192 ± 8.758	*p* = 0.113
Worse ear PTA	35.0000 ± 12.7037	29.2308 ± 14.49262	*p* = 0.272
MMSE	28.667 ± 1.3452	29.000 ± 1.7795	*p* = 0.578
**AEFs and MMNm**	**HQ**	**Ctrl**	***p* value**
Case	14	13	
Sex	M7, F7	M4, F9	*p* = 0.310
Age	67.29 ± 7.447	69.69 ± 9.295	*p* = 0.463
Better ear PTA	25.7143 ± 8.459	28.9523 ± 15.2227	*p* = 0.498
Worse ear PTA	36.6071 ± 15.6849	35.6731 ± 16.2469	*p* = 0.880
MMSE	29.143 ± 0.9493	29.077 ± 1.7059	*p* = 0.901
**AEFs and MMNm**	**HF-HQ**	**HF-Ctrl**	***p* value**
Case	10	8	
Sex	M6, F4	M3, F5	*p* = 0.343
Age	68.0 ± 8.246	68.25 ± 8.311	*p* = 0.950
Better ear PTA	23.875 ± 8.92581	21.7188 ± 10.0209	*p* = 0.636
Worse ear PTA	30.8750 ± 1.54175	29.0625 ± 14.86171	*p* = 0.766
MMSE	29.000 ± 0.9428	29.500 ± 1.4142	*p* = 0.382

**Table 3 biology-13-01021-t003:** Gap detection test (GDT) and fast speech test (FST). Data show mean ± SD. HQ: IH sound therapy group, Ctrl: control, HF-HQ: high-frequency hearing loss HQ, HF-Ctrl: high-frequency hearing loss control.

GDT (ms)		Pre			Post		
Better ear	HQ	6.6 ± 6.61			4.6 ± 1.56		
	Ctrl	6.11 ± 6.70			4.84 ± 3.69		
	HF-HQ	4.8 ± 2.17			4.27 ± 1.61		
	HF-Ctrl	4.17 ± 0.99			3.67 ± 0.75		
Worse ear	HQ	6.5 ± 6.48			4.72 ± 1.77		
	Ctrl	6.42 ± 6.79			4.63 ± 2.43		
	HF-HQ	4.93 ± 1.61			4.93 ± 1.44		
	HF-Ctrl	4.33 ± 1.97			3.83 ± 2.07		
**FST (%)**		**Pre**			**Post**		
Better ear	Group-speed	Beginning	Middle	End	Beginning	Middle	End
	HQ-1	86.84 ± 19.94	88.15 ± 24.45	88.94 ± 23.48	90 ± 12.35	90.52 ± 16.05	91.57 ± 13.48
	HQ-1.5	83.68 ± 19.25	80.52 ± 26.00	84.73 ± 26.72	86.84 ± 17.33	84.21 ± 22.02	88.42 ± 19.19
	HQ-2	64.73 ± 26.28	62.28 ± 30.96	67.36 ± 29.17	67.36 ± 23.19	63.85 ± 25.47	72.10 ± 24.29
	Ctrl-1	91.75 ± 9.390	88.5 ± 14.23	92 ± 12.88	94 ± 7	93 ± 8.276	94.5 ± 5.678
	Ctrl-1.5	87 ± 13.91	82.5 ± 17.64	85 ± 13.32	88.75 ± 10.23	86 ± 13.19	87.5 ± 10.18
	Ctrl-2	62.75 ± 18.87	53.5 ± 21.97	66 ± 20.40	64.66 ± 16.46	59.5 ± 23.28	69.25 ± 16.45
	HF-HQ-1	91.33 ± 10.07	90.33 ± 13.47	91 ± 14.51	93.66 ± 7.630	94.66 ± 6.699	95.66 ± 5.120
	HF-HQ-1.5	87 ± 14.58	83 ± 19.21	87.66 ± 13.02	90.33 ± 9.392	89.66 ± 10.56	89.33 ± 10.14
	HF-HQ-2	64.66 ± 20.61	58.33 ± 22.70	68.66 ± 22.17	68.55 ± 16.40	65.66 ± 21.97	74 ± 15.62
	HF-Ctrl-1	95 ± 4.564	97.91 ± 4.310	97.91 ± 3.796	93.75 ± 7.395	94.16 ± 11.14	95.83 ± 6.066
	HF-Ctrl-1.5	92.91 ± 5.187	91.66 ± 8.249	95.41 ± 7.204	89.58 ± 10.49	90.41 ± 14.78	92.91 ± 11.44
	HF-Ctrl-2	77.08 ± 13.91	74.86 ± 21.08	80.83 ± 13.20	75 ± 17.67	71.52 ± 22.43	77.5 ± 18.76
Worse ear	HQ-1	84.5 ± 15.48	81.5 ± 22.08	81.5 ± 21.45	91.05 ± 11.07	86.96 ± 14.21	90.52 ± 9.986
	HQ-1.5	72.75 ± 21.47	69.75 ± 26.52	69.5 ± 28.80	76.75 ± 25.01	72.25 ± 27.45	75.75 ± 26.32
	HQ-2	51.75 ± 24.61	44.75 ± 27.94	51 ± 29.56	52.66 ± 27.87	49.25 ± 30.17	55.5 ± 29.06
	Ctrl-1	84.47 ± 16.29	80.78 ± 25.81	81.84 ± 22.37	87.63 ± 17.27	89.21 ± 18.22	91.57 ± 13.48
	Ctrl-1.5	78.42 ± 22.42	75.78 ± 27.01	76.05 ± 30.41	81.84 ± 23.23	78.94 ± 29.36	82.63 ± 25.41
	Ctrl-2	57.36 ± 29.44	54.47 ± 30.38	61.31 ± 30.81	61.31 ± 28.96	62.36 ± 28.02	70 ± 28.09
	HF-HQ-1	86 ± 15.93	84 ± 21.46	84 ± 21.30	93.66 ± 7.180	90 ± 9.128	92.66 ± 6.548
	HF-HQ-1.5	75.66 ± 21.59	75.33 ± 24.45	74 ± 27.70	84.33 ± 10.93	82.66 ± 12.76	84 ± 14.28
	HF-HQ-2	54.66 ± 22.98	49.66 ± 28.77	55.66 ± 29.14	60.22 ± 20.42	57.66 ± 25.15	61.66 ± 22.92
	HF-Ctrl-1	91.25 ± 6.495	93.33 ± 9.204	91.66 ± 10.27	90 ± 15.54	90.83 ± 17.89	95.83 ± 6.066
	HF-Ctrl-1.5	86.66 ± 13.74	88.75 ± 12.93	90 ± 15.13	85 ± 21.60	84.58 ± 27.49	87.5 ± 21.65
	HF-Ctrl-2	70.41 ± 22.21	66.66 ± 25.52	72.91 ± 19.94	69.58 ± 26.65	67.91 ± 27.03	75.41 ± 24.95

**Table 4 biology-13-01021-t004:** The percentage of subjects showing improvement in N1m-P2m in the left STG at a stimulation sound level of 55 dB from the better hearing ear and MMNm amplitude in the left STG; (improvement subjects/total subjects). The HQ and HF-HQ groups exhibited a higher improvement rate compared to the Ctrl and HF-Ctrl groups. lt N1m-P2m improvement: improvement in inter-peak amplitude of N1m-P2m in left STG at a stimulation sound level of 55 dB from the better hearing ear, lt MMNm improvement: improvement in MMNm amplitude in the left STG.

	Lt N1m-P2m Improvement	Lt MMNm Improvement	Improvement in Either	Improvement in Both
HQ	86% (12/14)	79% (11/14)	93% (13/14)	71% (10/14)
Ctrl	46% (6/13)	46% (6/13)	69% (9/13)	23% (3/13)
HF-HQ	80% (8/10)	80% (8/10)	90% (9/10)	70% (7/10)
HF-Ctrl	38% (3/8)	38% (3/8)	63% (5/8)	13% (1/8)

**Table 5 biology-13-01021-t005:** LME model for the SINT with S/N +10 in the better hearing ear in the IH sound therapy group. From the fixed-effect summary, IH sound therapy and hearing ability under the threshold level of 70 dB at 10,000 Hz indicated significant differences, (*p* < 0.05 and *p* < 0.01), but inter-peak amplitude of N1m-P2m in the left STG and MMNm amplitude in the left STG did not show any significance. Over 70 dB at 10,000 Hz: hearing ability under the threshold level of 70 dB at 10,000 Hz; lt N1m-P2m amplitude: inter-peak amplitude of N1m-P2m in the left STG at a stimulation sound level of 55 dB from the better hearing ear. lt MMNm amplitude: MMNm amplitude in the left STG.

Fixed Effect	Estimate	Std. Error	Df	T Value	Pr(>|t|)		*p* Value
(Intercept)	53.70	115.27	7.15	0.47	0.66		
IH sound therapy	9.27	4.17	15.22	2.22	0.04		*p* < 0.01
Age	−0.47	0.50	7.32	−0.94	0.38		
PTA	0.41	0.52	7.33	0.79	0.45		
Over 70 dB at 10,000 Hz	−27.82	7.96	7.56	−3.50	0.01		*p* < 0.001
MMSE	1.97	4.31	7.04	0.46	0.66		
Lt N1m−P2m amplitude	−0.40	0.34	12.58	−1.19	0.26		
Lt MMNm amplitude	0.08	0.40	18.76	0.20	0.85		
**Correlation of fixed effect**	**(Intrecept)**	**IH sound therapy**	**Age**	**PTA**	**Over 70 dB at 10,000 Hz**	**MMSE**	**Lt N1m−P2m amplitude**
IH sound therapy	0.054						
Age	−0.017	−0.067					
PTA	0.39	0.103	0.286				
Over 70 dB at 10,000 Hz	0.072	−0.037	0.142	−0.24			
MMSE	−0.947	−0.035	−0.291	−0.525	−0.101		
Lt N1m−P2m amplitude	−0.085	−0.312	−0.001	−0.12	−0.136	0.066	
Lt MMNm amplitude	0.109	0.275	−0.199	0.17	−0.257	−0.03	0.176

## Data Availability

The data presented in this study are available upon reasonable request from the corresponding author.

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
