# Peer review of "Intelligibility Sound Therapy Enhances the Ability of Speech-in-Noise Perception and Pre-Perceptual Neurophysiological Response"

_biology, 2024, doi:10.3390/biology13121021_

Round 1

Reviewer 1 Report

Comments and Suggestions for Authors

In this manuscript, the authors reported the positive effect of intelligible sound therapy for aural rehabilitation. Using the IH sound collector, which emphasizes high-frequency sounds, the participants exhibited an improvement in speech-in-noise perception measured by AEFs and MMNm results after 35 days of sound therapy compared to the cohort with the normal sound collector. Overall, aligned with the previous publication from this group, this result may support the improvement of sound therapy using an intelligible-hearing sound collector. The data is straightforward, and the interpretation is fair.

Here, I have certain comments:

1.    Line 146, for this cohort, the authors may need to show the hearing threshold at each frequency. I assume most of them showed over 30 dB at high frequency. Please confirm.

2.    Table 1, the authors should also include the full name of all abbreviations, eg, PTA, MMSE, HQ, etc.

3.    Table 1, p value, did the authors mean = rather than <? Please confirm.

4.    In Table 1, the authors may need to specifically show the high-frequency hearing threshold. Since the IH speaker emphasizes the high tone, is there any difference between HQ and Ctrl groups of their high frequency hearing threshold?

5.    Figure 2, 6, seems like the Ctrl pre-therapy group always showed a higher accuracy rate compared to the HQ pre-therapy group. Is there any significant difference between these two groups? It is important because the authors use the above two groups as the baseline for the evaluation of therapy. If so, please state it in the discussion part.

6.    Line 27, ‘mismatch’ no need to capitalize.

7.    I suggest changing Figure 1 to Table 1 since the questionnaire is a table rather than a figure.

Author Response

  1. Line 146, for this cohort, the authors may need to show the hearing threshold at each frequency. I assume most of them showed over 30 dB at high frequency. Please confirm.

Thank you for giving us the suggestions. We agree with your suggestion. Therfore we add the figure showing the audiogram.

  1. Table 1, the authors should also include the full name of all abbreviations, eg, PTA, MMSE, HQ, etc.

Thank you for giving us the suggestions. We agree with that and we revised the table adding all abbreviations.

  1. Table 1, p value, did the authors mean = rather than <? Please confirm.

Thank you for giving us the suggestions. We agree with that and we revised p value in table 1.

  1. In Table 1, the authors may need to specifically show the high-frequency hearing threshold. Since the IH speaker emphasizes the high tone, is there any difference between HQ and Ctrl groups of their high frequency hearing threshold?    

Thank you for giving us the suggestions. We agree with the suggestion and add the figure showing the audiogram.

  1. Figure 2, 6, seems like the Ctrl pre-therapy group always showed a higher accuracy rate compared to the HQ pre-therapy group. Is there any significant difference between these two groups? It is important because the authors use the above two groups as the baseline for the evaluation of therapy. If so, please state it in the discussion part.

Thank you for giving us the suggestions. We agree with that and revised the limitation.

  1. Line 27, ‘mismatch’ no need to capitalize.

Thank you for giving us the suggestions. We revised the capitalize.

  1. I suggest changing Figure 1 to Table 1 since the questionnaire is a table rather than a figure.

Thank you for giving us the suggestions. We change the name from Figure 1 to Table 1

Reviewer 2 Report

Comments and Suggestions for Authors

This study investigated the effectiveness of Intelligible Hearing (IH) sound therapy, and its impact on neural processing of pure tone stimuli as well as a number of other behavioral measures, including pure tone averages from basic audiometry for the left and right ears, Mini-Mental State Examination (MMSE), GDT (Gap Detection Test), FST (Forward Speech Test), and SINT (Speech in Noise Test). The MEG tests involved basic evoked field responses (N1m and P2m) using a pure tone of 1000 Hz, and the MMNm (Mismatch Negativity) using 1000 Hz vs. 500 Hz pure tone stimuli in a passive oddball paradigm.

While the overall design adopts a comprehensive assessment of the hearing and cognitive measures as well as basic neural measures, there are a number of areas that need to be improved to interpret the results properly. Limitations need to be expanded and discussed. Below I list my major concerns.

1. It is problematic to use paired and independent t-tests for a study with multiple measures, such as repeated assessments of cognitive states, auditory processing tests, and neural measures. The reporting of p values in the tables looks unusual. If you set an alpha value criterion such as 0.05, you report the exact values such as p = 0.351 not p < 0.352. Please specify other key stats such as t value, the degree of freedom, etc. clearly. To reflect the study's design properly and to account for the complexities of the data, it would be beneficial to employ statistical methods that can manage multiple measures and consider the dependencies among them. Please consult with a statistician on how to process the data properly. There are at least two reasons against the simplistic t-test approach.

a. Each t-test increases the risk of Type I error (false positives). When conducting multiple tests, the probability of finding at least one significant result just by chance increases, potentially leading to misleading conclusions.

b. T-tests typically focus on mean differences, which may overlook the variability and patterns within the data. They do not consider the interactions between different factors (e.g., hearing thresholds, different sound levels, or ear stimulation).

Mixed repeated measures ANOVA or mixed effects models are needed to handle the complex data structure and account for the treatment effects and individual differences (as a random effect variable). The  mixed effects model beneficial for longitudinal data and can model repeated measures effectively. Posthoc tests can be used if there are interaction effects. Please use correction for multiple comparisons.

2. More detailed descriptions of the various tests can be helpful. The PTA and MMSE measures did not reveal significant changes between the experimental and control group and before and after therapy according to the researchers' t-test results. Similarly, no significant differences or changes were observed in the GDT and FST tests. In the SINT test, speech perception in noisy environments was assessed using Japanese words mixed with speech-spectrum noise at various S/N ratios (-15, -10, -5, 0, +5, +10 dB). Significant improvement was noted only at the S/N ratio of +10 dB in the experimental group (HF-HQ), while the control group did not show any significant changes. Please note that more sophisticated statistical analysis is needed here. Tests involving multiple comparisons need to be corrected.

4. In the MEG measures, the evoked responses used a 1000 Hz tone presented at sound levels of 50, 55, and 60 dB HL. Significant improvement in the N1m-P2m amplitude was observed in the left superior temporal gyrus (STG) for the experimental groups (HQ and HF-HQ) after sound therapy, while no significant changes were noted in the control groups. In the oddball paradigm, a significant increase in MMNm amplitude was observed in the left STG for both HQ and HF-HQ groups after therapy (p<0.01 for HQ and p<0.05 for HF-HQ). The MMNm results were interpreted to support the notion that IH sound therapy positively influenced automatic auditory processing. Again, more sophisticated statistical models are needed to handle the data structure here.

5. Given the different auditory processing abilities assessed, it would be beneficial to discuss how these behavioral and neural measures interrelate and their implications for therapy effectiveness. In a mixed effects model, predictor variables such as PTA, cognitive measures, etc. can all be included. But the small sample size may limit the statistical power and generalizability of the findings, especially for complex analyses.

6. The study does not specify the long-term effects of IH sound therapy. Thus there is a limitation in its conclusions about sustainability over time.

7. It is important to consider how age, hearing loss and cognitive decline contribute to the MMNm and N1m, and P2m measures. There a are number of related studies by Koerner and Zhang et al. that I recommend the researchers to check out. For example,

Koerner, T.K., & Zhang, Y. (2018). Differential effects of hearing impairment and age on electrophysiological and behavioral measures of speech in noise. Hearing Research, 370, 130-142.

Koerner, T. K., & Zhang, Y. (2017). Application of linear mixed-effects models in human neuroscience research: A comparison with Pearson correlation in two auditory electrophysiology studies. Brain Sciences, 7, 26.

Koerner, T. K., Zhang, Y., Nelson, P., Wang, B., & Zou, H. (2016). Neural indices of phonemic discrimination and sentence-level speech intelligibility in quiet and noise: A mismatch negativity study. Hearing Research, 339, 40-49

Comments on the Quality of English Language

The manuscript can benefit from some editing to improve clarity. Complex sentences can be broken down for easier reading.

Author Response

  1. It is problematic to use paired and independent t-tests for a study with multiple measures, such as repeated assessments of cognitive states, auditory processing tests, and neural measures. The reporting of p values in the tables looks unusual. If you set an alpha value criterion such as 0.05, you report the exact values such as p = 0.351 not p < 0.352. Please specify other key stats such as t value, the degree of freedom, etc. clearly. To reflect the study's design properly and to account for the complexities of the data, it would be beneficial to employ statistical methods that can manage multiple measures and consider the dependencies among them. Please consult with a statistician on how to process the data properly. There are at least two reasons against the simplistic t-test approach.
  2. Each t-test increases the risk of Type I error (false positives). When conducting multiple tests, the probability of finding at least one significant result just by chance increases, potentially leading to misleading conclusions.
  3. T-tests typically focus on mean differences, which may overlook the variability and patterns within the data. They do not consider the interactions between different factors (e.g., hearing thresholds, different sound levels, or ear stimulation).

Mixed repeated measures ANOVA or mixed effects models are needed to handle the complex data structure and account for the treatment effects and individual differences (as a random effect variable). The mixed effects model beneficial for longitudinal data and can model repeated measures effectively. Posthoc tests can be used if there are interaction effects. Please use correction for multiple comparisons.

Thank you for your advice. We agree with your comments and corrected the statistical analysis adding the effect of multiple comparisons with the Bonferroni approarch.

  1. More detailed descriptions of the various tests can be helpful. The PTA and MMSE measures did not reveal significant changes between the experimental and control group and before and after therapy according to the researchers' t-test results. Similarly, no significant differences or changes were observed in the GDT and FST tests. In the SINT test, speech perception in noisy environments was assessed using Japanese words mixed with speech-spectrum noise at various S/N ratios (-15, -10, -5, 0, +5, +10 dB). Significant improvement was noted only at the S/N ratio of +10 dB in the experimental group (HF-HQ), while the control group did not show any significant changes. Please note that more sophisticated statistical analysis is needed here. Tests involving multiple comparisons need to be corrected.

Thank you for your advice. We agree with your comments and corrected the statistical analysis adding the effect of multiple comparisons with Bonferroni approach. In MMSE, the data was collected only before the sound therapy so we could not compare between pre and post therapy. PTA was not changed in each group between pre and post therapy. For more detail, we add the data of PTA before the therapy. GDT and FST tests results also added as table 3.

  1. In the MEG measures, the evoked responses used a 1000 Hz tone presented at sound levels of 50, 55, and 60 dB HL. Significant improvement in the N1m-P2m amplitude was observed in the left superior temporal gyrus (STG) for the experimental groups (HQ and HF-HQ) after sound therapy, while no significant changes were noted in the control groups. In the oddball paradigm, a significant increase in MMNm amplitude was observed in the left STG for both HQ and HF-HQ groups after therapy (p<0.01 for HQ and p<0.05 for HF-HQ). The MMNm results were interpreted to support the notion that IH sound therapy positively influenced automatic auditory processing. Again, more sophisticated statistical models are needed to handle the data structure here.

Thank you for your advice. We agree with your comment and corrected the statistical analysis adding the effect of multiple comparisons with the Bonferroni approach.

  1. Given the different auditory processing abilities assessed, it would be beneficial to discuss how these behavioral and neural measures interrelate and their implications for therapy effectiveness. In a mixed effects model, predictor variables such as PTA, cognitive measures, etc. can all be included. But the small sample size may limit the statistical power and generalizability of the findings, especially for complex analyses.

Thank you for giving us the suggestions. As you mentioned, we performed the mixed effects model but the sample size for age, cognitive measures etc was small and could not measure the effects.

  6. The study does not specify the long-term effects of IH sound therapy. Thus there is a limitation in its conclusions about sustainability over time.

Thank you for giving us the suggestions. We added the limitation about it.

 7. It is important to consider how age, hearing loss and cognitive decline contribute to the MMNm and N1m, and P2m measures. There a are number of related studies by Koerner and Zhang et al. that I recommend the researchers to check out.

Thank you for giving us the suggestions. We added the limitation about it.

Round 2

Reviewer 1 Report

Comments and Suggestions for Authors

The authors have improved the manuscript according to the comments. I endorse the publication of this paper.

Author Response

The authors have improved the manuscript according to the comments. I endorse the publication of this paper.

Thank you for reviewing our manuscript. We appreciate your thoughtful suggestions.

Reviewer 2 Report

Comments and Suggestions for Authors

Your responses indicate that you adopted more sophisticated statistical approaches in your revision. But you did not do report any details of the multivariate models. Instead, you kept all your t-test results with minor modifications with Bonferroni correction and p-value reporting. Did you actually do the Bonferroni correction, which should change the p-values in the tables? In any case, the t-test approach is not appropriate for your analysis and reporting given the longitudinal design involving two subject groups, and multiple measures of behavioral and neural data. 

Your analysis did not address the relationships or interactions among the behavioral measures, nor did you report any connections between brain data and behavioral outcomes.  Age and cognitive measures may exhibit limited variance due to the subject selection criteria. However, hearing thresholds demonstrated a wide range in the PTA measure. Figure 2 clearly shows an effect of the signal-to-noise (S/N) ratio on the SINT measures, which is probably affected by hearing threshold as assessed in the PTA measure, yet statistical tests for these relationships are absent. 

Did the pure-tone average (PTA) predict any of the SINT measures? It would be strange that PTA has nothing to do with SINT performance. Moreover, did the MMN or N1/P2 data correlate with the SINT measures before or after training? If there is no relationship between MMN and the SINT measures, you actually showed that the neural plasticity observed in the training as measured in MMN has nothing to do with behavioral performance in speech--in-noise tests. That should be discussed in great detail. Similarly, I am also curious about this question: How could the presentation levels of 50, 55, and 60 dB HL not affect the N1m and P2m responses? You did not report any statistical test results on these modulating factors. Ignoring these relationships in a report involving multivariate measures within a longitudinal design raises significant concerns.

Although your primary focus was on the pre- versus post-training comparison, the reliance on multiple t-tests overlooked the interrelationships among the various measures. For instance, the S/N ratio systematically influenced SINT accuracy across both subject groups, as indicated in Figure 2. How did the PTA factor into this? Did it affect SINT measures at different S/N ratios? Could the neural measures such as MMN provide insights into the SINT outcomes before and after training?

If you found no statistical relationships among any measures, it raises questions about the credibility of your findings. Multivariate analyses using mixed-effects models are not merely recommended; they are essential for exploring the connections among age, PTA, cognitive measures, SINT accuracy, and MEG data. SINT accuracy is a key behavioral measure here, often used in studies (like those by Koerner and Zhang) that investigate the relationship between neural metrics, such as MMN, and speech-in-noise performance in individuals with hearing loss. Surprisingly, despite a wealth of behavioral measures reported, there are no relationships or interactions revealed or discussed among them.

While significant pre- and post-training changes are noteworthy, understanding what drives these changes and accounts for individual differences is crucial for uncovering the mechanisms of neural plasticity and behavioral changes. As I noted in my previous review, using t-tests is not a suitable approach given your experimental design, which includes a control and experimental group and measures taken before and after training for both neural and behavioral data.

Comments on the Quality of English Language

Complex sentences can be simplified to improve readability. 

Author Response

You did not do report any details of the multivariate models. Instead, you kept all your t-test results with minor modifications with Bonferroni correction and p-value reporting. Did you actually do the Bonferroni correction, which should change the p-values in the tables? In any case, the t-test approach is not appropriate for your analysis and reporting given the longitudinal design involving two subject groups, and multiple measures of behavioral and neural data. Your analysis did not address the relationships or interactions among the behavioral measures, nor did you report any connections between brain data and behavioral outcomes.  Age and cognitive measures may exhibit limited variance due to the subject selection criteria. However, hearing thresholds demonstrated a wide range in the PTA measure. Figure 2 clearly shows an effect of the signal-to-noise (S/N) ratio on the SINT measures, which is probably affected by hearing threshold as assessed in the PTA measure, yet statistical tests for these relationships are absent. Did the pure-tone average (PTA) predict any of the SINT measures? It would be strange that PTA has nothing to do with SINT performance. Moreover, did the MMN or N1/P2 data correlate with the SINT measures before or after training? If there is no relationship between MMN and the SINT measures, you actually showed that the neural plasticity observed in the training as measured in MMN has nothing to do with behavioral performance in speech-in-noise tests. That should be discussed in great detail.

Thank you for your advice. We performed the analysis using an LME model, adding random effects for MMN, N1m-P2m, age, PTA, and the group with over 75dB at 10000Hz to evaluate the improved percentage of SINT+10. However, the model fit was singular, as indicated by the warning message. We removed each item one by one and found that none of the random effects contributed to the model. This suggests that all possible random effects are biased. Consequently, we combined some items for further analysis in the LME model, focusing only on HQ therapy, as shown in Table 2. Based on your advice, we have added the results showing the improvement percentages of N1m-P2m and MMNs after sound therapy.

Similarly, I am also curious about this question: How could the presentation levels of 50, 55, and 60 dB HL not affect the N1m and P2m responses? You did not report any statistical test results on these modulating factors. Ignoring these relationships in a report involving multivariate measures within a longitudinal design raises significant concerns.

Thank you for your question. The sound level refers to the stimulus sound level. At this point, we focused on the level of stimulus sound for further analysis and found that 55dB was the most suitable for further analysis.

Although your primary focus was on the pre- versus post-training comparison, the reliance on multiple t-tests overlooked the interrelationships among the various measures. For instance, the S/N ratio systematically influenced SINT accuracy across both subject groups, as indicated in Figure 2. How did the PTA factor into this? Did it affect SINT measures at different S/N ratios? Could the neural measures such as MMN provide insights into the SINT outcomes

Thank you for your question. We performed the analysis using an LME model, adding random effects for MMN, N1m-P2m, age, PTA, and the group with over 75dB at 10000Hz to evaluate the improved percentage of SINT+10. However, the model fit was singular, as indicated by the warning message, and we found that all the possible random effects were biased. Consequently, we could not detect any correlation between PTA and SINT results. Instead, we focused on the factor of PTA using HQ and HF-HQ.

If you found no statistical relationships among any measures, it raises questions about the credibility of your findings. Multivariate analyses using mixed-effects models are not merely recommended; they are essential for exploring the connections among age, PTA, cognitive measures, SINT accuracy, and MEG data. SINT accuracy is a key behavioral measure here, often used in studies (like those by Koerner and Zhang) that investigate the relationship between neural metrics, such as MMN, and speech-in-noise performance in individuals with hearing loss. Surprisingly, despite a wealth of behavioral measures reported, there are no relationships or interactions revealed or discussed among them.

Thank you for your advice. We could not perform the analysis using an LME model with random effects for MMN, N1m-P2m, age, PTA, and the group with over 75dB at 10000Hz due to bias. However, your advice is important for further analysis, so we have mentioned it in the limitations section.

Round 3

Reviewer 2 Report

Comments and Suggestions for Authors

Thank you for the responses and revision. However, your descriptions in the main text and in the responses indicate that you did not implement LME properly. "Linear mixed-effects models was performed to determine whether these objective neural measures were able to predict “age”, “MMSE score”, and “HQ sound therapy”. In this analysis, participant was included as a “by-subject” random effect in the linear mixed-effect model and “HQ sound therapy” and “pre- and post-therapy” was set as fixed effects."  Further, you described that all the neural measures, PTA, age, etc. were treated as random effect variables in your analysis. There is a complete misunderstanding of what LME analysis is. In fact, all conventional parametric test methods including the t-tests, ANOVA, etc. are just special cases of LME. They should be and can be directly tested within LME. Please read this paper and consult with a statistician seriously to get all analyses done properly starting with normality checking first for each variable. If a key variable is not normally distributed, consider using generalized linear mixed effect models. Be extremely cautiously with the use of random effect variables. I recommend hiring a statistician to help address the problems and not relying on SPSS to solve the problem. As the statistical analysis was not done properly, I cannot recommend acceptance. 

Yu, Z., Guindani, M., Grieco, S. F., Chen, L., Holmes, T. C., & Xu, X. (2022). Beyond t test and ANOVA: applications of mixed-effects models for more rigorous statistical analysis in neuroscience research. Neuron110(1), 21–35. https://doi.org/10.1016/j.neuron.2021.10.030

Author Response

Thank you for the responses and revision. However, your descriptions in the main text and in the responses indicate that you did not implement LME properly. "Linear mixed-effects models was performed to determine whether these objective neural measures were able to predict “age”, “MMSE score”, and “HQ sound therapy”. In this analysis, participant was included as a “by-subject” random effect in the linear mixed-effect model and “HQ sound therapy” and “pre- and post-therapy” was set as fixed effects."  Further, you described that all the neural measures, PTA, age, etc. were treated as random effect variables in your analysis. There is a complete misunderstanding of what LME analysis is. In fact, all conventional parametric test methods including the t-tests, ANOVA, etc. are just special cases of LME. They should be and can be directly tested within LME. Please read this paper and consult with a statistician seriously to get all analyses done properly starting with normality checking first for each variable. If a key variable is not normally distributed, consider using generalized linear mixed effect models. Be extremely cautiously with the use of random effect variables. I recommend hiring a statistician to help address the problems and not relying on SPSS to solve the problem. As the statistical analysis was not done properly, I cannot recommend acceptance. 

Yu, Z., Guindani, M., Grieco, S. F., Chen, L., Holmes, T. C., & Xu, X. (2022). Beyond t test and ANOVA: applications of mixed-effects models for more rigorous statistical analysis in neuroscience research. Neuron110(1), 21–35. https://doi.org/10.1016/j.neuron.2021.10.030

We appreciate the feedback regarding our statistical approach. After consulting with statistical specialists, they initially identified multiple regression analysis as the most appropriate method for our study and mentioned how to do the LME model. We added the LME model showing the association between each item as fixed effects and changed the manuscript.